# Ballistocardial Signal-Based Personal Identification Using Deep Learning for the Non-Invasive and Non-Restrictive Monitoring of Vital Signs

**DOI:** 10.3390/s24082527

**Published:** 2024-04-15

**Authors:** Karin Takahashi, Hitoshi Ueno

**Affiliations:** Faculty of Information Design, Tokyo Information Design Professional University, Edogawa-ku, Tokyo 132-0034, Japan; takahashi@tid.ac.jp

**Keywords:** monitoring system, piezoelectric sensor, bio-signal, personal identification

## Abstract

Owing to accelerated societal aging, the prevalence of elderly individuals experiencing solitary or sudden death at home has increased. Therefore, herein, we aimed to develop a monitoring system that utilizes piezoelectric sensors for the non-invasive and non-restrictive monitoring of vital signs, including the heart rate and respiration, to detect changes in the health status of several elderly individuals. A ballistocardiogram with a piezoelectric sensor was tested using seven individuals. The frequency spectra of the biosignals acquired from the piezoelectric sensors exhibited multiple peaks corresponding to the harmonics originating from the heartbeat. We aimed for individual identification based on the shapes of these peaks as the recognition criteria. The results of individual identification using deep learning techniques revealed good identification proficiency. Altogether, the monitoring system integrated with piezoelectric sensors showed good potential as a personal identification system for identifying individuals with abnormal biological signals.

## 1. Introduction

With the increasing aging trends in the global population, a discernible surge has been observed in the prevalence of conditions affecting elderly individuals, including bedridden states and solitary living conditions. Sudden deaths of elderly individuals are not limited to single-person households but also extend to those with multiple residents. Considering the accelerated societal aging, the incidence of individuals passing away at home could follow an upward trajectory.

Various elderly monitoring sensors have been developed to ensure the safety of elderly individuals living independently. These sensors acquire signals using methods such as video and audio recording [1], infrared sensors [2,3], door open/close sensors [4], a flow sensor [5,6], or the use of positioning sensing mats [7,8]. They monitor the condition of the elderly and provide necessary support. Additionally, these sensor systems are becoming more efficiently operated through the utilization of wireless sensors via the internet [9] and wearable [10,11,12,13] sensors. Furthermore, systems utilizing machine learning have been developed to detect anomalies such as falls [14] and abnormal postures [15] in the elderly, enabling remote support.

In monitoring systems, sensors can broadly be categorized into those that monitor and acquire information about human actions and those that monitor and acquire vital signs. Among monitoring systems, acquiring vital signs facilitates understanding of health conditions, making it applicable in nursing, caregiving, and medical fields. Monitoring sensors that acquire vital signs are primarily divided into two types: restraint sensors that require direct attachment to the body, and non-restraint sensors that function without direct contact with the body. Restraint sensors acquire physiological information from the elderly and quickly detect sudden changes in health conditions, thus being used in many nursing and caregiving monitoring systems. Therefore, the assessment of changes in health status necessitates a time frame exceeding several hours. In essence, a challenge persists in conventional monitoring systems due to the absence of a method that is both non-restraining and capable of real-time detection of physiological signals.

We have consistently proposed a senior monitoring system as a solution to this challenge from a conventional standpoint [16]. The system monitors the heartbeat and respiration of the elderly in a non-restrictive manner while they are inside their residence, which is achieved by installing numerous piezoelectric sensors throughout the household. This allows the elderly to rapidly detect changes in their health status without experiencing any inconvenience. However, a challenge with this system arises when multiple individuals reside in the same household, making it difficult to identify specific individuals. Therefore, we aim to construct an elderly monitoring system that utilizes cardiac pulsation signals acquired from piezoelectric sensors, enabling non-intrusive and non-restrictive monitoring of vital signs. Additionally, to address the challenges in households with multiple residents, our goal is to incorporate individual authentication features based on cardiac pulsation signals. The importance of personal identification through ballistocardial signals can also extend to the field of nursing [17].

We devised a methodology for individual identification based on the distinctive shape of the peak in the frequency spectra of the ballistocardial signals, which were proposed to comprise a novel identification criterion [16,18]. Our approach differs from other monitoring systems in that it can identify subjects not considered in many other monitoring systems and can be used to perform subject identification without using unconstrained sensors. This study thus presents a promising and highly effective method for monitoring systems within multiple households, as it can be applied to surveillance systems that combine individual identification with minimal stress on the subjects, making it easier to identify individuals without causing stress to the monitored subjects.

Owing to the fluctuating intensity of ballistocardial signals obtained from an individual on an hourly and daily basis, straightforward identification through simple calculations is challenging. In this study, we explored the feasibility of individual identification based solely on cardiac pulsation signal data by utilizing deep neural networks, which are useful for extracting personal characteristics from the constantly changing spectrum. Additionally, we scrutinized the capability of our approach, as an initial step in employing machine learning for personal identification, to discern individuals within paired datasets by classifying the data derived from different individuals.

## 2. Materials and Methods

### 2.1. Ballistocardial Measuring Device

A ballistocardiogram (BCG) is a diagram illustrating minute vibrations manifesting on the surface of the body owing to the oscillations in the blood vessels, such as arteries, correlated with heart pulsations. Vibrations originating from the heartbeat allow for the measurement of BCG-derived beat-to-beat intervals (BBI) using a piezoelectric sensor. This BBI corresponds to the respiration rate intervals of the electrocardiogram, demonstrating almost identical values and facilitating heart rate calculations [19,20]. Additionally, fluctuations in arterial blood pressure contain signals caused by respiration, facilitating the computation of the respiration rate from these observations [21].

A piezoelectric sensor is a transducer that generates an electric charge in response to fluctuations in applied pressure. The electric charge is amplified using a sensor amplifier, acquiring a voltage value corresponding to the pressure change. Herein, a methodology was employed to quantify the aforementioned action using a custom-made sensor (Health Sensing Co., Ltd., Tokyo, Japan), which was installed on a chair (Figure 1a), where the individuals were directed to sit. When individuals are sitting or standing on a piezoelectric sensor, the pressure exerted on the sensor changes due to the vibration of the person’s arteries appearing on the surface of the body. Even when individuals are wearing clothing, vibrations can still be transmitted, thus obviating the need for sensors to be tightly affixed to their bodies (Figure 1b). The sensor signals were obtained at 100 Hz, amplified, analog-to-digital converted, and displayed on a computer screen (Figure 1c).

### 2.2. Measurement Procedures

The subjects walked to the measurement site, but to avoid the instability of heart rate due to physical activity, they rested quietly for 10 min upon arrival before conducting measurements for over 200 s.

The noise signals arising from minor body movements were detected, even in a seated position. Figure 2a shows a typical example of the obtained sensor signal. The noise signals were interspersed at various points within the data obtained during the 200 s measurement. Figure 2b shows an enlarged view of the stable region (range of 126 s to 138 s) of the example signal (Figure 1a). The illustrated peaks correspond to the heartbeat and the vibrational patterns associated with breathing. In this study, the original data shown in Figure 2 were collected from seven individuals and used for the analysis.

### 2.3. Workflow for Training Datasets

The BCG signals of seven individuals (age range approximately 20–65 years) were analyzed. The BCG signals were obtained for 200 s or more for each participant sitting on the chair equipped with a piezoelectric sensor. For the entirety of the measurement range, evaluations were conducted utilizing signals generated when the subjects were seated, and the time domain excluding periods affected by saturation was employed for analysis. Within this framework, precise time intervals (in approximately 10 s increments) were delineated. The training dataset for each individual was augmented using these signals. Because the stability of the data obtained differed depending on the participant, we determined and verified the stability range for each individual. Therefore, the measurement time used for the training data and the temporal timing of test data differed slightly. The workflow of the training data augmentation and inference datasets is illustrated in Figure 3. First, a measurement region with stable data was selected based on the on/off signals of the sensor. Second, multiple spectra were generated by clipping the spectra, with a data length of 5 s, from a randomly selected starting point. Third, the generated spectra were fast Fourier transformed (FFT), wherein a high-pass filter was employed to substitute components within the frequency range of 0 to 0.1 Hz with an intensity value of 0. Furthermore, a low-pass filter was subsequently applied to eliminate high-frequency components exceeding 15 Hz. Finally, each spectrum was normalized such that the maximum value was 1.

Herein, 5000 pieces of training data were generated for each individual, and the data length inputted into the model was 150, through random data extraction and low-pass filtering. The number of datasets was set at 5000, within a range where excessive duplication of data is avoided. The 150 input data points were obtained by performing FFT processing on a signal acquired at a sampling frequency of 100 Hz, and then applying a frequency filter to obtain the remaining measurement points. The inference data were generated from measurement regions that were not used for training, and 10–20 sets were prepared for each individual. The datasets for each individual and the regions of the training and test datasets in its spectrum are shown in Figure 4. The individual data of seven people were labeled P1 to P7.

### 2.4. Workflow for Training Models

The deep neural network architecture comprised three dense layers and was designed to classify two sets of individual data (Figure 5). The initial layer with 150 input features was connected to a fully connected layer containing 32 neurons. The output from this layer was subjected to the rectified linear unit (ReLU) activation function. Subsequently, a hidden layer consisting of another fully connected layer with 16 neurons, also employing ReLU activation, was present. The final layer, which served as the output layer, comprised two units and utilized the sigmoid activation function to produce a probability distribution, thereby scaling the output to a range between 0 and 1 for two classes. The employed loss function L was the binary cross-entropy, and optimization was performed using the Adam optimizer with a learning rate of 1.0 × $10^{-2}$. The models were trained for a maximum of five epochs.

Individual test data were inputted into the trained model, and the resulting probabilities were displayed for each of the two classifications. If the probability of belonging to the same classification category as the training data was high (i.e., a probability of 0.5 or more), it was labeled as a positive result. Conversely, if the probability of being classified as another person’s data was high (i.e., not belonging to the expected category), it was labeled as a negative result.

## 3. Results and Discussion

### 3.1. Processing Time

The training time for 5000 datasets was approximately 10 s. The typical loss value decreased from 0.2 to 0.0002 after five epochs (Figure 6). The inference processing time for each dataset was 20 ms.

### 3.2. Accuracy for Personal Identification

We investigated the ability of our approach to identify individuals based on the ballistocardial signals using data from seven individuals, labeled P1 to P7. The 5000 training datasets and 10–20 inference datasets were created for each individual (Section 2.3). In total, two sets were selected from the data of P1–P7, resulting in 21 combinations for binary classification (7 × 6/2). Table 1 presents the percentage of positive results.

Of the 21 combinations, 19 (approximately 90%) yielded positive results. The classification cases for P1 and P5 (negative results) and for P6 and P7 (positive results) are illustrated in Figure 7 and Figure 8, respectively. Both figures show subsets of training and inference data.

In almost all cases, the accuracy of correct answers was commendable, suggesting a nearly successful classification. However, it is crucial to scrutinize the cases in which the accuracy rate did not reach 50%, such as the P1–P5 pair. We identified two potential reasons for this classification inability, which are as follows:(1)Frequency spectral similarity: The variation in the dissimilarity between the frequency spectral similarities of individuals makes it difficult to distinguish individuals who exhibit similarity. To enhance the classification, it might be feasible to incorporate secondary information sources, such as the heart rate, rather than relying solely on the frequency spectra.(2)Failure to extract frequency-spectrum components (measurement instability): It is conceivable that the frequency-spectrum components cannot be reliably extracted. Occasionally, appropriate ballistrocardial signals were not obtained, such as when the subject had just repositioned themselves or in cases of a slight movement. In such cases, identification can be improved by continually scrutinizing the classification information based on the spectra of the preceding and subsequent timeframes. Persistent differences in the identifier over a certain duration could signify the presence of another person assuming the seat. However, momentary variations suggest that an appropriate BCG signal was not detected, prompting the need for identifier adjustment. This iterative approach could enhance the personal identification performance.

The classification of P1 and P5 represented the most challenging case in this study (Figure 7). The accuracy rate of the inference result of P5 was 80%, whereas that of P1 was 25% (Table 1), indicating that 75% of the P5 test data were classified as P1. In contrast, the classifications of P6 and P7 (Figure 8) showed that all inference instances were successfully categorized.

The classification cases for P1 and P5 were believed to involve a combination of the conditions described in (1) and (2). When examining the spectral shapes that could not be classified (Figure 7), the frequencies of the main peaks of P1 and P5 were similar. Extending the measurement time to extend the training data resulted in some improvements in the classification results of P1 and P5. However, although increasing the extraction time of the training data enhanced the resolution of the FFT spectrum, it made the spectrum susceptible to the effects of body movements during measurements, which is a limitation. Adjusting the extraction time on a case-by-case basis could thus be effective.

The inference results of P5 (Figure 7) showed instances where only a portion failed, which could be attributed to the reason mentioned in (2). However, considering the real-world operational scenario, where the continuous examination of classification information based on spectral data before and after a specified time is essential, such cases are not problematic. This is because individuals are unlikely to swap positions within 5 s (the time needed to obtain one spectrum in this study). The cases in which the classification was easily achievable involved differences in the frequencies of the main peaks (Figure 8). Even if there was variability in the intensity and frequency of the spectrum, such cases could be classified without any issues in this study.

Previous research struggled to effectively perform individual identification using statistical methods. This was because it required consideration of multivariate variables, and relying solely on representative variables proved inadequate for individual identification. However, incorporating deep learning in this study enabled the extraction of hidden features, demonstrating a high capability for individual identification. This study’s results showed that among 21 analysis patterns, 19 patterns were capable of identifying individuals with an accuracy of over 50%, and it was demonstrated that individuals could be identified with 90% accuracy. This study represents a novel approach to individual identification, acquiring vital signs non-intrusively an non-restrictively, making it a significant advancement in the field.

The vibration signals detected by piezoelectric sensors are believed to originate from the pulsation of arterial vessels due to changes in blood pressure. It is estimated that individual differences arise from factors such as the arterial shape, cardiac movement, and blood viscosity of the individual. Therefore, the accuracy of individual identification is not expected to reach the precision of fingerprint authentication. However, by advancing the identification method based on piezoelectric sensor signals, it is believed that accuracy at the level of intra-family identification within a household can be achieved.

## 4. Conclusions

Herein, we developed a monitoring system by integrating sheet-shaped piezoelectric sensors for household use and showed its potential to evolve into a personal identification system adept at discerning individuals with abnormal biological signals, even within households accommodating multiple residents. The present study exclusively showcased the identification proficiency for two individuals; future studies are aimed at broadening this capacity to encompass four to five individuals, aligning it with the demographic norm of a single household, accommodating >10 individuals, and addressing scenarios involving several residents in communal facilities, such as nursing homes. Altogether, the approach shown in this study represents a systematic exploration of identity verification functions and has the potential to enhance the efficiency and precision of personal authentication.

## Figures and Tables

**Figure 1 sensors-24-02527-f001:**
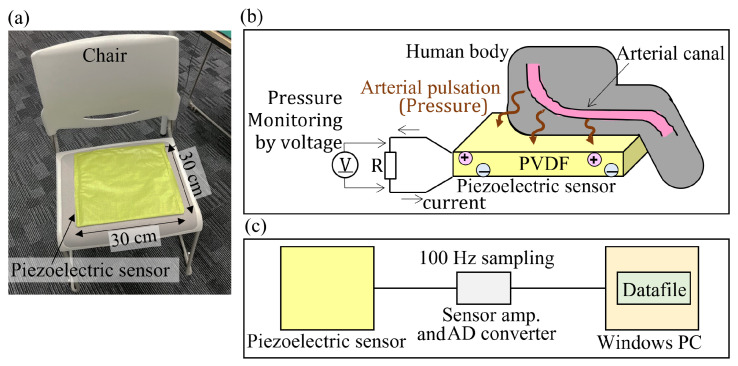
Ballistocardial measuring device. (**a**) Piezoelectric sensor installed on the chair. (**b**) The mechanism of acquiring heart pulsation signals. (**c**) Sensor system configuration. PVDF, polyvinylidene diFluoride; AD, analog-to-digital; PC, personal computer.

**Figure 2 sensors-24-02527-f002:**
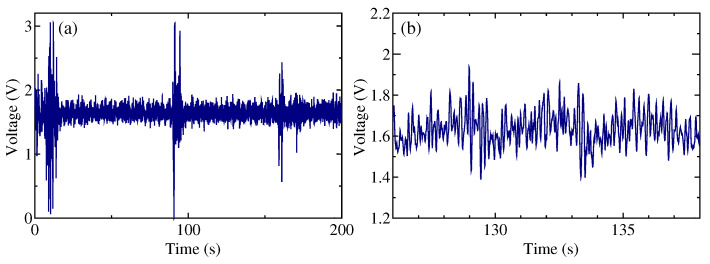
Typical sensor signals. (**a**) Typical example of the sensor signal for an individual. (**b**) Enlarged view of the stable region of (**a**) in the range of 126 s to 138 s.

**Figure 3 sensors-24-02527-f003:**
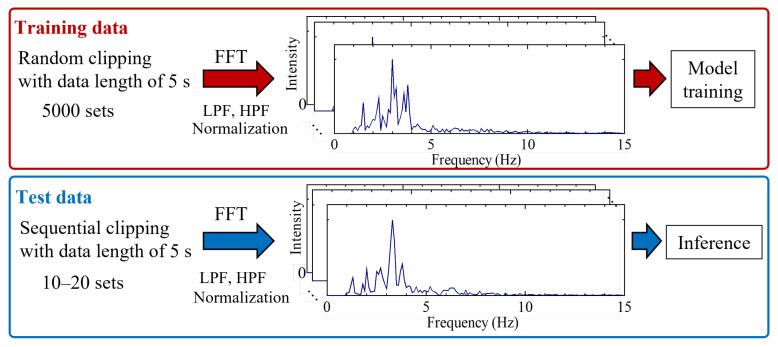
Workflow of training data augmentation and inference datasets. FFT, fast Fourier transformation; LPF, low-pass filtering; HPF, high-pass filtering.

**Figure 4 sensors-24-02527-f004:**
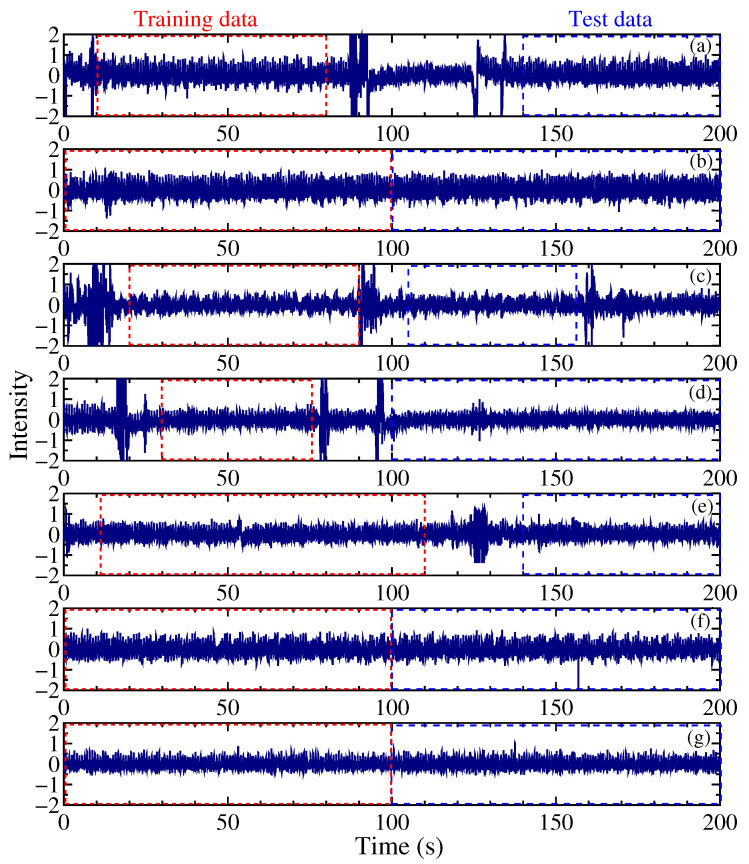
Ballistocardial signals for (**a**) P1, (**b**) P2, (**c**) P3, (**d**) P4, (**e**) P5, (**f**) P6, and (**g**) P7. The ranges of training data and test data are shown with a red dotted line and blue short dashes, respectively. The vertical axis is normalized to ±1 in the usage area.

**Figure 5 sensors-24-02527-f005:**
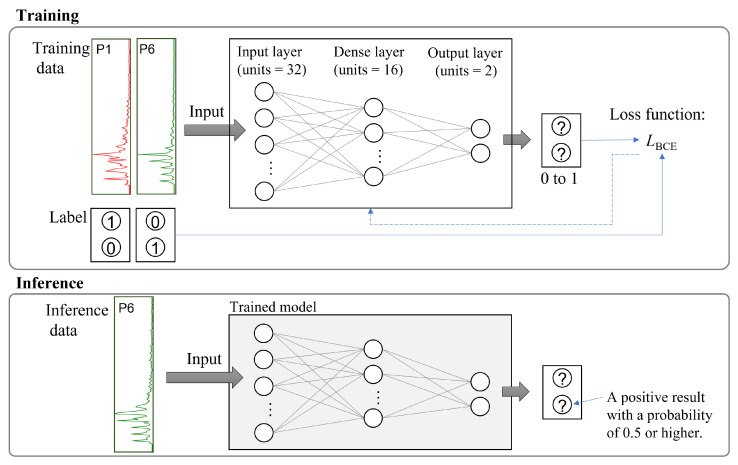
Workflow of training model and inference data.

**Figure 6 sensors-24-02527-f006:**
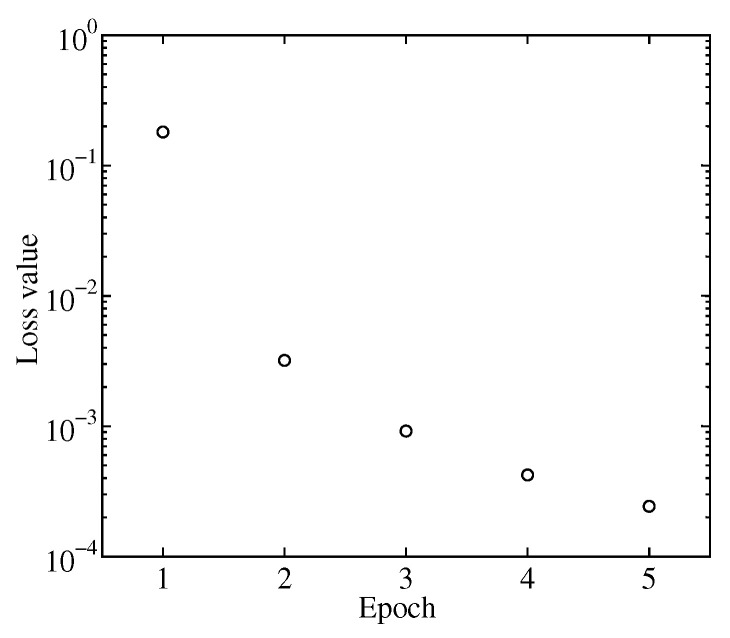
Loss value for each epoch.

**Figure 7 sensors-24-02527-f007:**
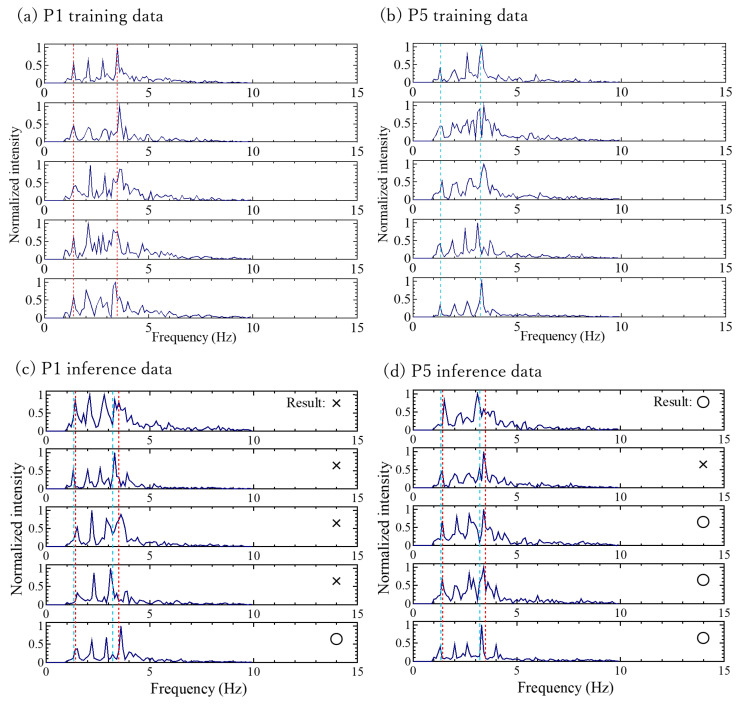
Subsets of the training data of (**a**) P1 and (**b**) P5. Subsets of the inference data and classification results of (**c**) P1 and (**d**) P5. The red dotted line represents the main peak position of P1, and the blue dashed line represents the main peak position of P5. Inference data are shown with the frequency on the horizontal axis and normalized intensity on the vertical axis.

**Figure 8 sensors-24-02527-f008:**
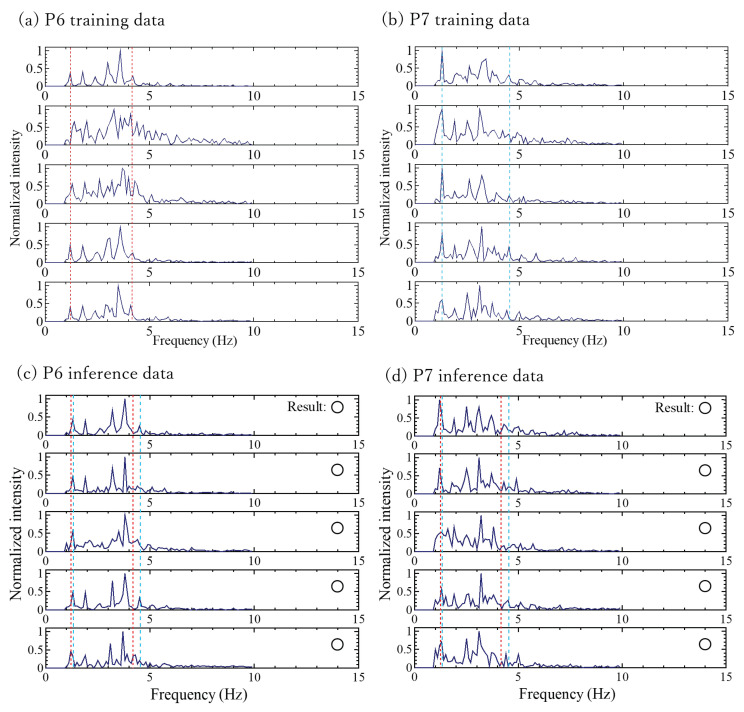
Subsets of the training data of (**a**) P6 and (**b**) P7. Subsets of the inference data and classification results of (**c**) P6 and (**d**) P7. The red dotted line represents the main peak position of P6, and the blue dashed line represents the main peak position of P7. Inference data are shown with frequency on the horizontal axis and normalized intensity on the vertical axis.

**Table 1 sensors-24-02527-t001:** Accuracy rate (%) of inference results.

Train A, Train B	Inference A	Inference B
P1, P2	67%	100%
P1, P3	67%	100%
P1, P4	83%	90%
P1, P5	25%	80%
P1, P6	100%	100%
P1, P7	83%	100%
P2, P3	100%	100%
P2, P4	100%	40%
P2, P5	100%	70%
P2, P6	100%	100%
P2, P7	100%	100%
P3, P4	100%	90%
P3, P5	100%	80%
P3, P6	100%	100%
P3, P7	100%	95%
P4, P5	55%	80%
P4, P6	85%	100%
P4, P7	85%	100%
P5, P6	100%	100%
P5, P7	100%	100%
P6, P7	100%	100%

Train A/B, the pair of training data sets for A/B. A and B correspond to the mentioned P1–P7 pair.

## Data Availability

The data supporting the findings of this study are available upon request from the corresponding author.

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
