# Peer review of "Ballistocardial Signal-Based Personal Identification Using Deep Learning for the Non-Invasive and Non-Restrictive Monitoring of Vital Signs"

_sensors, 2024, doi:10.3390/s24082527_

Round 1
Reviewer 1 Report
Comments and Suggestions for Authors
I found this contribution of K. Takahashi and H Ueno very interesting looking at the usefulness of the proposed sensor and the quality of results enabled. This contribution is, however, improvable if authors can give more information:
1) On how raw data was collected: a particular part of the day, after exercise, after 15 min. seat, etc.
2) Why the neural network was settled with so many neurons (32) and an exceeding 150 input features? What were them? Frequency, intensity, frequency range, mean peak-to-peak amplitude, peak slope?
3) How do authors now the time window for the samples feedind was optimal?
4) How inference % changes for different larger or shorter training set numbers?
Minor changes:
Line 20: systems
Line 117: linear
Reviewer 2 Report
Comments and Suggestions for Authors
This article proposes a monitoring system for identifying individuals with abnormal biological signals using piezoelectric sensors. The system is designed to monitor elderly individuals living independently and detect sudden deaths. The monitoring sensors can be categorized into two types: restraining sensors and non-restraining sensors. The system uses a ballistocardiogram (BCG) measuring device and a workflow for training datasets to extract frequency-spectrum components and identify individuals based on the shapes of these peaks. The results of individual identification using deep learning techniques revealed good identification proficiency. The monitoring system integrated with piezoelectric sensors showed good potential as a personal identification system for identifying individuals with abnormal biological signals. Overall, the article is a valuable work and is recommended for publication. Although, the authors should polish the following issues:
(1) Further references should be reviewed in the introduction section. Only 12 papers have been used to illustrate and introduce the background of the work. At the same time, references 1-7 appeared in one sentence. It is not a convictive introduction.
(2) A discussion section is needed to discuss the innovation point of this work.
(3) The detection principle of the piezo-sensor should be illustrated.
Comments on the Quality of English Language
Good enough but not very good
Round 2
Reviewer 1 Report
Comments and Suggestions for Authors
All issues of the original manuscript were in my opinion conveniently addressed.